# Reproducibility of strength performance and strength-endurance profiles: A test-retest study

**Benedikt Mitter** ⓘ*, **Robert Csapo**◉, **Pascal Bauer**◉, **Harald Tschan**◉

Centre for Sport Science and University Sport, University of Vienna, Vienna, Austria

◉ These authors contributed equally to this work.
* benedikt.mitter@univie.ac.at

**Data Availability Statement:** All relevant data are within the manuscript and its Supporting Information files. Scripts used for the statistical analyses can be accessed using the DOI: https://doi.org/10.5281/zenodo.5840363.

## Abstract

The present study was designed to evaluate the test-retest consistency of repetition maximum tests at standardized relative loads and determine the robustness of strength-endurance profiles across test-retest trials. Twenty-four resistance-trained males and females (age, 27.4 ± 4.0 y; body mass, 77.2 ± 12.6 kg; relative bench press one-repetition maximum [1-RM], 1.19 ± 0.23 kg•kg$^{-1}$) were assessed for their 1-RM in the free-weight bench press. After 48 to 72 hours, they were tested for the maximum number of achievable repetitions at 90%, 80% and 70% of their 1-RM. A retest was completed for all assessments one week later. Gathered data were used to model the relationship between relative load and repetitions to failure with respect to individual trends using Bayesian multilevel modeling and applying four recently proposed model types. The maximum number of repetitions showed slightly better reliability at lower relative loads (ICC at 70% 1-RM = 0.86, 90% highest density interval: [0.71, 0.93]) compared to higher relative loads (ICC at 90% 1-RM = 0.65 [0.39, 0.83]), whereas the absolute agreement was slightly better at higher loads (SEM at 90% 1-RM = 0.7 repetitions [0.5, 0.9]; SEM at 70% 1-RM = 1.1 repetitions [0.8, 1.4]). The linear regression model and the 2-parameters exponential regression model revealed the most robust parameter estimates across test-retest trials. Results testify to good reproducibility of repetition maximum tests at standardized relative loads obtained over short periods of time. A complementary free-to-use web application was developed to help practitioners calculate strength-endurance profiles and build individual repetition maximum tables based on robust statistical models.

## Introduction

Dynamic strength endurance has previously been defined as the amount of concentric work an individual can produce in a cyclic or repetitive movement [1]. Assuming that the range of motion is approximately constant for each repetition of a given resistance training exercise, strength endurance can therefore be described by the number of repetitions performed to momentary failure (RTF) at a given load for a single sustained trial [1,2]. The evaluation of

**Funding:** Open access funding provided by University of Vienna.

**Competing interests:** The authors have declared that no competing interests exist.

strength endurance by means of a repetition maximum test (occasionally also called repetition endurance test) usually involves an exercise being performed to momentary failure at either a fixed absolute load, expressed in a unit of mass like kg or lbs, or a fixed relative load that has been normalized to the exercise-specific one-repetition maximum (1-RM). The concept is widely applied by coaches to guide resistance training programming [1,3,4]. However, given the fact that resistance training is usually carried out across a wider spectrum of loads, assessing the RTF an individual can execute at a single load only provides limited insight into a person's fatigue resistance. More meaningful insights into strength endurance could be obtained by studying the relationship between load and RTF (i.e., the individual "strength-endurance profile"). Additionally, knowledge of the mathematical relationship between the two variables could be used by practitioners to predict the load associated with a certain repetition maximum. This may be of particular interest for individuals seeking to control intensity of effort within a set [5] by prescribing a certain percentage of the maximum load that can be used for a given number of repetitions [6]. While other methods have been proposed to evaluate or control intensity of effort based on perceived effort or movement velocity [7], an approach using strength-endurance profiles might overcome certain limitations of these methods. Such limitations include inappropriate anchoring of perception [5], inaccurate subjective estimates of repetitions in reserve at lower intensity of effort [8] and dependency on technology to provide reliable feedback on movement velocity [9].

The relationship between load and RTF can be expressed through simple bivariate models. Thus far, research has proposed models that describe either a linear [10–12] or an exponential relationship [11–14]; usually, the respective model equations are then rearranged to predict the 1-RM from a repetition maximum test. However, studies conducted to test the validity of these equations often showed poor predictive accuracy, especially when the applied repetition maximum test was executed at loads allowing for 10 repetitions or more [3,12,13,15–17]. The poor validity may be related to substantial inter-individual differences in strength-endurance relationships that models not incorporating the responsible confounding factors fail to account for. Indeed, there is evidence that the amount of repetitions that can be performed at a given relative load, and hence the strength-endurance relationship, may depend on various factors, such as qualitative and quantitative training background [11,18–20], fiber type composition and the capillary density of involved muscles [21,22], exercise [12,19,23] and movement cadence [14,24]. A possible solution to overcome these challenges in modeling strength-endurance relationships has been proposed by Morton and colleagues [25] who introduced the idea of creating subject-specific models, thereby treating the individual person as the population of interest. For this purpose, the authors reformulated the critical power model originally proposed by Monod and Scherrer [26] such that it may be applied to isoinertial resistance training exercises. The resulting model has recently been referred to as critical load model and was originally presented as a non-linear function featuring three parameters [25,27].

While the individualized modeling approach may reduce variance resulting from uncontrolled confounding variables, such models are typically estimated from a limited number of available data due to the exhaustive nature of sets performed to failure [25,27]. Hence, the estimation of model parameters can be strongly affected by variability in test results, as single data points tend to have a larger influence on parameter estimates in small samples compared to large samples. Therefore, the robustness of individual strength-endurance models over short periods is crucial for their application in practice. The present study was designed to target two objectives: 1) to evaluate the consistency of the RTF at standardized relative loads and 2) to compare the reproducibility of four recently proposed models describing the individual strength-endurance relationship. A complementary, freely available web application will be provided to allow practitioners to easily calculate strength-endurance profiles based on a

model that can be considered sufficiently robust to help with the design and regulation of resistance training programs.

## Materials and methods

### Subjects

Fifteen resistance-trained men (age = 27.2 ± 3.3 yrs, body mass = 85.4 ± 7.9 kg, bench press 1-RM/body mass = 1.33 ± 0.11 kg•kg$^{-1}$) and nine resistance-trained women (age = 27.7 ± 5.2 y, body mass = 63.6 ± 3.3 kg, bench press 1-RM/body mass = 0.96 ± 0.17 kg•kg$^{-1}$) volunteered to be tested for the present study. In order to participate, subjects had to be between 18 and 40 years of age, free of illness and injury and have at least one year of training experience in the bench press exercise as well as a 1-RM corresponding to at least 1x body mass for men and 0.75x body mass for women, respectively. Prior to physical testing, participants were informed about the possible risks, had to complete a modified physical activity readiness questionnaire (PAR-Q) and sign an informed consent form. The study was designed in fulfillment of the ethical guidelines communicated in the Declaration of Helsinki and approved by the host institution's local ethical committee (no. 00461).

### Experimental approach

A test-retest design was used to determine the participants' maximum strength and strength-endurance at high loads in the free-weight bench press exercise on two occasions (T1 & T2) separated by one week (Fig 1). Maximum strength was assessed according to a progressive 1-RM test. Strength-endurance was assessed using repetition maximum tests at 90%, 80% and 70% of the 1-RM, respectively. In order to provide some rest, the 1-RM test and the repetition maximum tests were executed on two different days, separated by 48 to 72 hours, resulting in a total of four visits to the laboratory within 11 days. Importantly, the relative loads applied for the repetition maximum tests during the fourth visit were adjusted to the 1-RM achieved during the third visit. Consequently, a change in the 1-RM between T1 and T2 also implied a change in the absolute load used for the repetition maximum tests at T2, in order to ensure that trials were performed at 90%, 80% and 70% of the current 1-RM. All tests were completed at the same time of day.

### Procedures

On the first day, subjects completed preliminary health screening and filled in a physical activity form to evaluate their experience with the tested exercise. Body height and body mass were assessed using a stadiometer (Seca Model 217; SECA GmbH & Co. KG., Hamburg, Germany) and scale (Seca Model 877; SECA GmbH & Co. KG., Hamburg, Germany). Participants then completed a standardized warm-up consisting of cycling for 5 min at a constant power output of 1 W per kg body mass and a rotational velocity of 80 rpm on an ergometer (Kettler X1, Trisport, Huenenberg, Switzerland), followed by a brief dynamic upper body mobilization routine. Subsequently, they were familiarized with the standardized movement technique for the bench press: each subject had to lower the barbell onto two safety pins, which were individually adjusted to a height that would allow for up to 3 cm of vertical distance between the bottom barbell position and the subject's chest. An experienced staff member visually ensured that the barbell was placed on the safety pins without rebound, before giving the verbal command "Press!", signaling the subject to execute the concentric phase of the bench press at maximum voluntary velocity. Participants were required to maintain their hip, shoulders and head positioned on the bench and their feet placed on the floor during each set.

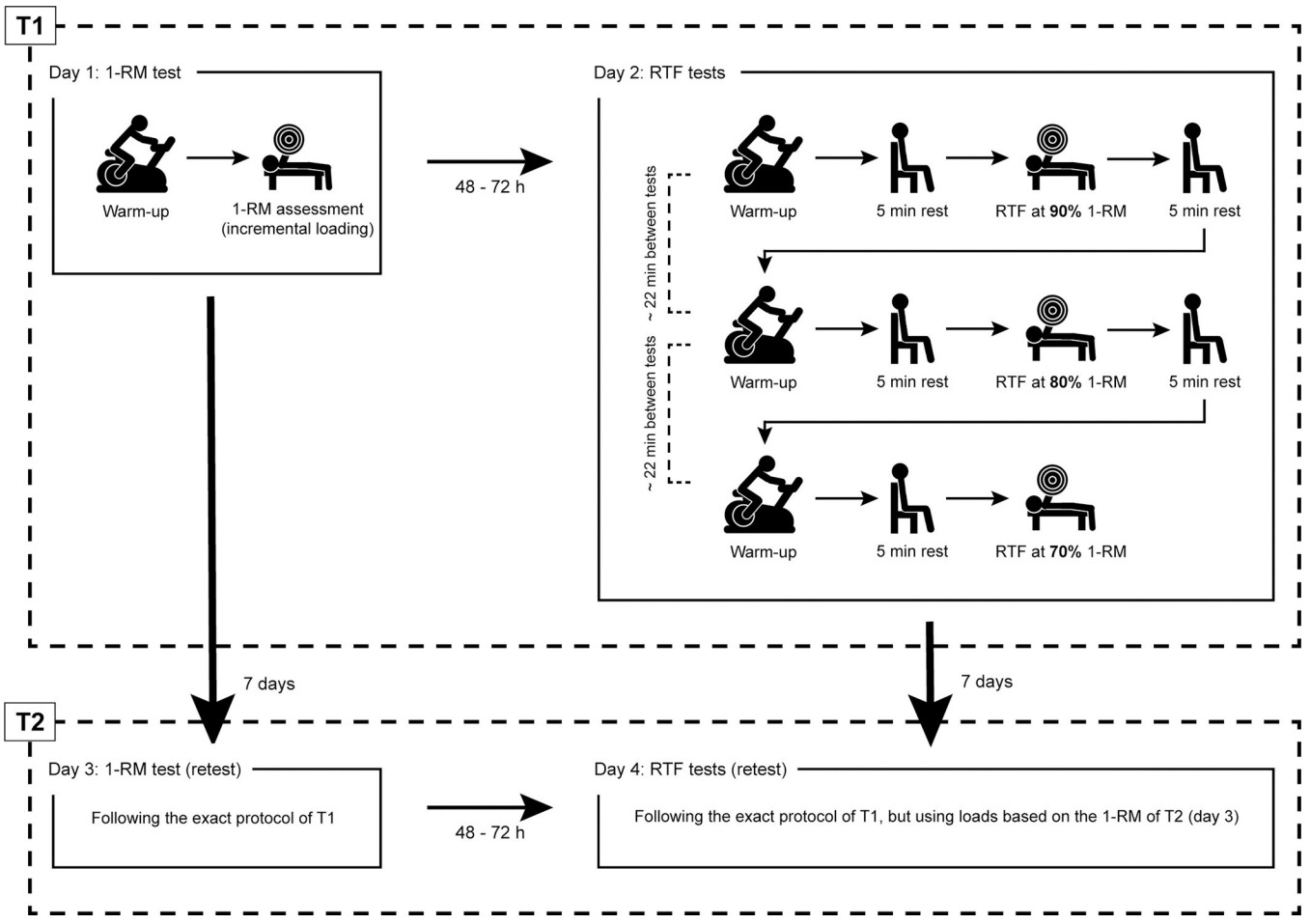

**Fig 1. Experimental design.** 1-RM, one-repetition maximum; RTF, repetitions performed to momentary failure.

Upon completion of the familiarization, participants were requested to estimate their 1-RM based on self-evaluation of their recent training performance. The subsequent 1-RM test featured a progressive loading pattern with the first five loads being fixed at 25%, 50%, 75%, 85% and 95% of the estimated 1-RM, while mean concentric barbell velocity was recorded with a linear position transducer (GymAware Power Tool, Kinetic Performance Technologies, Canberra, Australia). In the initial set, three repetitions were performed, followed by a 3-min break. Two repetitions were performed once the highest achieved velocity of the preceding set fell below 1.0 m/s, followed by a 4-min break and single repetitions were performed once it fell below 0.65 m/s, followed by a 5-min break. After successfully completing 95% of the estimated 1-RM, loads were increased individually to approximate the true 1-RM. The 1-RM was considered to be determined once a load increment of 2.5 kg from the preceding set would no longer allow the subject to complete the exercise across the full range of motion.

Repetition maximum tests were completed at 90%, 80% and 70% of the identified 1-RM, respectively, in the form of a single-visit protocol. Barbell loads were not randomized, but prescribed in a descending scheme, to minimize systematic effects of accumulated fatigue on the performance during subsequent sets [28]. In order to provide extended time for recovery in between repetition maximum tests, yet sustain warm-up effects during these periods, participants underwent the same general warm-up procedure that was used for the 1-RM test prior

to each set to failure. Additionally, they performed a specific warm-up including three repetitions at 25%, three repetitions at 50% and two repetitions at 75% 1-RM prior to each set to failure. A passive rest of 3 min was provided between warm-up sets and additional 5 min before and immediately after each set to failure. Due to this methodological structure (i.e., the standardized warm-up; the standardized passive rest before and after each set to failure), the repetition maximum tests were separated by approximately 22 min each. Criteria for movement execution were kept identical to those communicated for the 1-RM test. Participants were instructed to lower the barbell in a controlled fashion on each repetition, albeit not being prescribed a fixed movement cadence. Similar to the 1-RM test, participants had to await the verbal command of the staff member before initiating the concentric phase of a repetition, in order to avoid any rebound from the safety pins. The concentric phase of each repetition had to be performed at maximum intended velocity. A repetition maximum test was terminated once the participant was unable to complete another repetition across the full range of motion despite using maximal effort, suggesting that the point of momentary failure had been reached [5].

## Statistical analysis

**Reproducibility of performance measures.** Statistics were calculated following a Bayesian approach using weakly informative priors. To assess test-retest reliability of the 1-RM and RTF at 90%, 80% and 70% 1-RM, respectively, the following random-intercept mixed effects model was used:

$$P_{ij} \sim Normal(\mu + s_i + \Delta t D_j, \sigma_e^2) \tag{1}$$

In this model, $P_{ij}$ describes the analyzed performance measure as a dependent variable, $\mu$ describes the mean performance for T1, $s_i$ the random deviation from $\mu$ for subject $i$, $\Delta t$ the fixed effect of time (i.e., the systematic difference in performance between T2 and T1), $D_j$ a binary dummy variable for trial $j$, and $\sigma_e^2$ the variance of model residuals. The random effect parameter $s_i$ was considered to be sampled from a normal distribution with a mean of 0 and a variance of $\sigma_s^2$, as suggested by Baumgartner and colleagues [29]. Posterior distributions for each model parameter were sampled using the Hamiltonian Monte Carlo algorithm of the probabilistic programming language Stan [30] controlled through an R interface (*rstan* R package, version 2.21.2). Based on the resulting random-intercept models, relative consistency (reliability) of each performance measure was evaluated using the Intraclass Correlation Coefficient (ICC), which was estimated and interpreted as the proportion of total variance ($\sigma_s^2 + \sigma_e^2$) attributed to the variance among subjects ($\sigma_s^2$) [29]. Furthermore, absolute consistency (agreement) of performance was quantified using the Standard Error of Measurement (SEM = $\sigma_e$), Within-Subject Coefficient of Variation (WSCV = SEM / $\mu$) and Standard Error of Prediction (SEP = SD(1-ICC$^2$)$^{(1/2)}$) [29,31,32]. Posterior distributions of the statistics were summarized and interpreted according to the Maximum a Posteriori point estimate (MAP) and 90% Highest Density Interval (HDI) [33]. Effect directions supported by at least 90% of posterior probability were considered "clear" or "likely".

**Reproducibility of strength-endurance models.** To describe the relationship between relative load and RTF with respect to individual trends, four previously proposed model types were expressed according to a multilevel (mixed effects) structure:

$$Lin : \; load_i \sim Normal(\boldsymbol{a}_i + \boldsymbol{b}_i RTF_i, \sigma^2) \tag{2}$$

$$Ex2 : \; load_i \sim Normal(\boldsymbol{a_i}e^{(\boldsymbol{b_i}RTF_i)}, \sigma^2) \tag{3}$$

$$Ex3: \ load_i \sim Normal(\boldsymbol{c}_i + \boldsymbol{a}_i e^{(\boldsymbol{b}_i RTF_i)}, \sigma^2) \tag{4}$$

$$Crit: \ load_i \sim Normal(\boldsymbol{L'}_i/(RTF_i - \boldsymbol{k}_i) + \boldsymbol{CL}_i, \sigma^2) \tag{5}$$

Eq 2 (Lin) models the relationship as a linear regression. Eqs 3 (Ex2) and 4 (Ex3) both describe exponential regression models, where Ex3 follows the structure of a commonly proposed 3-parameters model [11,12,14] and Ex2 constitutes a simplified 2-parameters version without the additive parameter $\boldsymbol{c}_i$ [13]. Eq 5 (Crit) presents the previously described critical load model adapted for relative load as dependent variable, using the original parameter labels $\boldsymbol{L'}$, $\boldsymbol{k}$ and $\boldsymbol{CL}$ [25,27]. To evaluate how much parameter estimates for Eqs 2 to 5 differ between T1 and T2, a change effect was added for each of the abovementioned subject-level parameters. For example, the parameter expression $\boldsymbol{a}_i$ was extended to $(\boldsymbol{a}_{i\ +}\Delta\boldsymbol{a}_i\,D_j)$, where $\boldsymbol{a}_i$ reflects the target parameter at T1, $\Delta\boldsymbol{a}_i$ reflects the change effect (difference) of the target parameter between T2 and T1 and $D_j$ reflects a binary dummy variable for trial $j$. Importantly, all of the abovementioned parameters were modeled as random effects that were free to vary across subjects. The multilevel structure was realized by sampling subject-level parameters and change effects from multivariate normal distributions, applying covariance matrices to account for possible correlations among subject-level parameters and change effects, respectively. Further details on models and prior selection are provided online (Supporting information 1 in S1 Appendix).

A posterior predictive distribution was calculated by drawing random samples from the respective group-level (fixed effects) distribution of each change effect and the draws were standardized to the scale of the associated model parameter at T1. The resulting posterior predictive distributions were summarized and compared to a threshold for acceptable differences that was set at ±0.6, reflecting a small or trivial standardized change of the parameter [34]. Change effects were also expressed as a percentage of the group-level mean of the associated model parameter at T1 to facilitate the interpretation of parameters that are exceptionally homogeneous across subjects.

## Results

The variability of 1-RM performance as well as the RTF performed at 90%, 80% and 70% 1-RM is shown in Fig 2. On average, there was an increase in performance between T1 and T2 ($\Delta t$), the 90% HDI suggesting a small systematic increase of the 1-RM, the RTF at 80% 1-RM and the RTF at 70% 1-RM. Regarding relative consistency of performance, the 1-RM yielded nearly perfect reliability, with the ICC being close to 1. The RTF, on the other hand, showed a trend for higher relative consistency at lower loads, although the difference between load conditions was not statistically clear at the 90% credibility level. Analysis of absolute consistency revealed the SEM for the 1-RM to be likely less than 2.2 kg (90th percentile). Concerning the RTF performed at submaximal loads, the SEM was likely less than 1 repetition at 90% and 80% 1-RM, and likely less than 1.5 repetitions at 70% 1-RM. Subject performance and consistency statistics are summarized in Table 1.

Posterior predictive distributions of subject-level model parameters at T1 and T2 are summarized in Table 2. Moreover, posterior predictive distributions for standardized change effects are shown in Fig 3. The critical load model revealed a systematic positive change effect for $\boldsymbol{L'}$ [p ($\Delta\boldsymbol{L'}_i > 0$ | data) > 99.9%] and systematic negative change effects for $\boldsymbol{k}$ [p ($\Delta\boldsymbol{k}_i < 0$ | data) > 99.9%] and $\boldsymbol{CL}$ [p ($\Delta\boldsymbol{CL}_i < 0$ | data) = 97.3%]. Similarly, the 3-parameters exponential model showed a systematic positive change effect for $\boldsymbol{c}$ [p ($\Delta\boldsymbol{c}_i > 0$ | data) = 99.7%] and systematic negative change effects for $\boldsymbol{a}$ [p ($\Delta\boldsymbol{a}_i < 0$ | data) = 99.6%] and $\boldsymbol{b}$ [p ($\Delta\boldsymbol{b}_i < 0$ | data) =

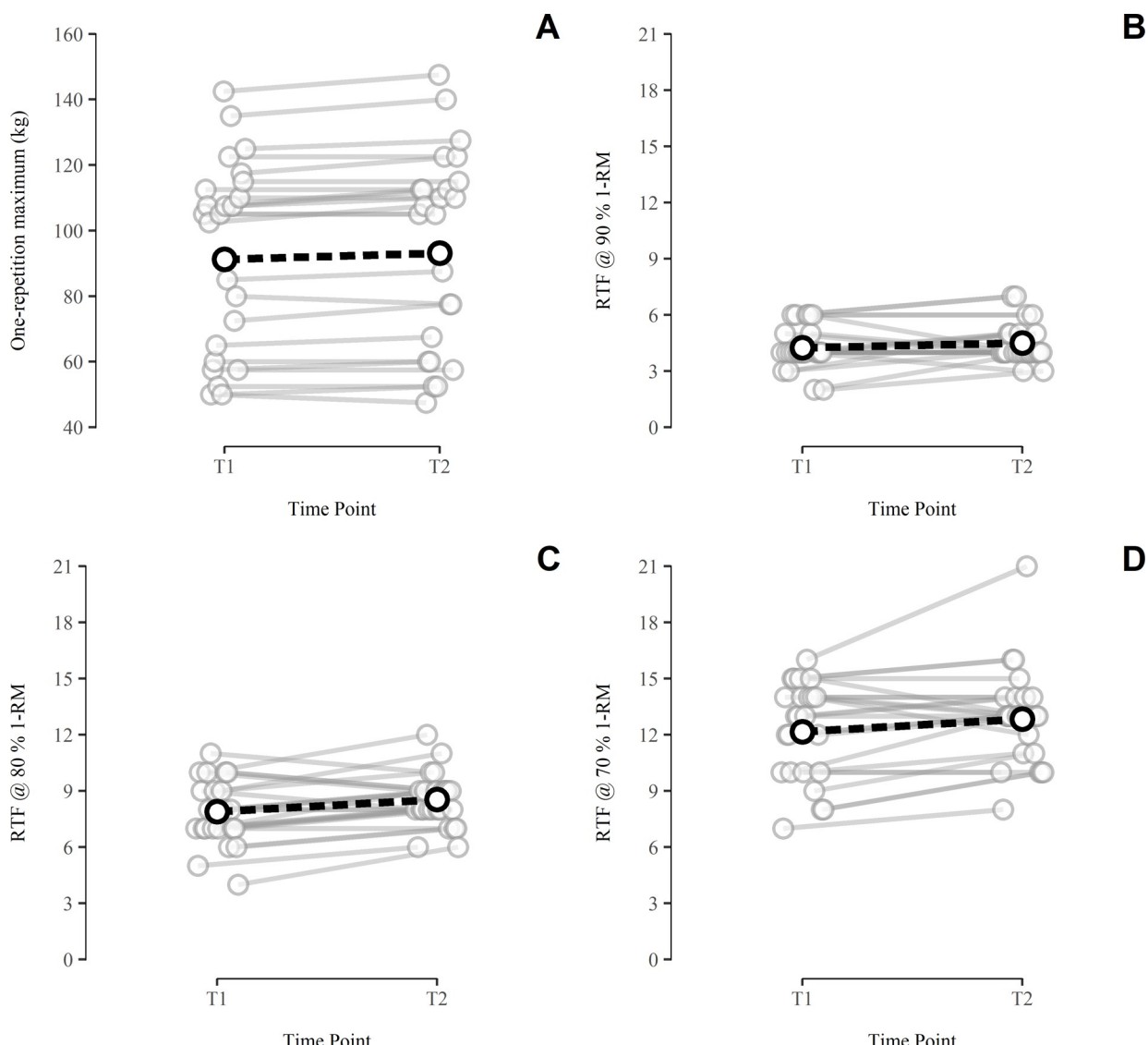

**Fig 2. Variability of strength performance in the bench press.** A, one-repetition maximum (1-RM); B, repetitions performed to momentary failure (RTF) at 90% 1-RM; C, RTF at 80% 1-RM; D, RTF at 70% 1-RM; grey circles, data points (jittered illustration); black circles, group means; solid grey lines, individual performance changes; dashed black lines, systematic performance changes ($\Delta t$).

96.4%]. None of the remaining models' parameters resulted in a clear positive or negative change at the 90% credibility level. No model parameter resulted in a clearly small or trivial change at the chosen credibility level and threshold for acceptable differences. However, the slope parameter of the linear model (**b**) and the curvature parameter of the 2-parameters exponential model (**b**) indicated a probability of >80% for the change effect to be small or trivial. Furthermore, both intercept parameters (**a**) of the linear model and the 2-parameters exponential model indicated relative change effects close to 0 (Table 3).

## Discussion

The present study was designed to address two objectives: first, we evaluated the reliability and agreement of RTF performed at 90, 80 and 70% 1-RM in the bench press exercise. Second, we

**Table 1. Consistency statistics for strength performance in the bench press.**

|  | 1-RM (kg) | RTF at 90% 1-RM (n) | RTF at 80% 1-RM (n) | RTF at 70% 1-RM (n) |
|---|---|---|---|---|
| **Performance** |  |  |  |  |
| T1 | 93.5 ± 28.9 | 4.2 ± 1.2 | 7.8 ± 1.7 | 12.2 ± 2.6 |
| T2 | 95.4 ± 29.9 | 4.5 ± 1.1 | 8.5 ± 1.4 | 12.9 ± 2.6 |
| Δt | 1.9 [1.0, 2.7] | 0.2 [-0.1, 0.6] | 0.7 [0.3, 1.0] | 0.7 [0.2, 1.2] |
| **Absolute consistency** |  |  |  |  |
| SEM | 1.7 [1.4, 2.3] | 0.7 [0.5, 0.9] | 0.7 [0.6, 1.4] | 1.1 [0.8, 1.4] |
| WSCV (%) | 1.8 [1.4, 2.5] | 15.9 [12.3, 21.3] | 9.2 [7.2, 12.2] | 8.8 [6.9, 11.8] |
| SEP | 2.3 [1.6, 3.3] | 0.9 [0.7, 1.1] | 0.9 [0.7, 1.9] | 1.4 [1.0, 1.9] |
| **Relative consistency** |  |  |  |  |
| ICC | 1.00 [0.99, 1.00] | 0.65 [0.39, 0.83] | 0.82 [0.64, 0.93] | 0.86 [0.71, 0.93] |

Sample data are presented as mean ± standard deviation.

Statistics are presented as Maximum a Posteriori estimate [90% Highest Density Interval].

1-RM, one-repetition maximum; Δt, fixed effect of time; ICC, interclass correlation coefficient; RTF, repetitions performed to momentary failure; SEM, standard error of measurement; SEP, standard error of prediction; T1, baseline test; T2, retest; WSCV, within-subject coefficient of variation.

aimed to analyze the reproducibility of four different models representing the individual strength-endurance relationship to identify which ones provide the most robust parameter estimates. Test-retest analysis of performance indicated very good reproducibility of the 1-RM and the RTF at high relative loads in the bench press exercise. The linear regression and the 2-parameters exponential regression yielded the most robust parameter estimates across the investigated models of the strength-endurance relationship.

The 1-RM revealed both very high relative and absolute consistency. In particular, the SEM for the 1-RM was found to be likely less than the smallest load increment applied during the 1-RM assessment in the present study (2.5 kg). These findings correspond to previous research reporting excellent reliability of 1-RM performance in the bench press exercise [4,35,36]. Similarly, the RTF at 90, 80 and 70% 1-RM revealed high absolute consistency, the SEM likely being less than 1.5 repetitions at 70% 1-RM, and less than 1 repetition at 90% and 80% 1-RM. Posterior distribution analysis revealed no systematic differences of SEM between RTF

**Table 2. Summary of posterior predictive distributions of absolute parameter values during test (T1) and retest (T2).**

| Model | Parameter | T1 | T2 | $\Delta x_i$ (T2 −T1) |
|---|---|---|---|---|
| Lin | a | 101.5 [100.3, 102.5] | 101.9 [100.4, 103.4] | 0.4 [-0.8, 1.6] |
|  | b | -2.73 [-3.77, -1.68] | -2.56 [-3.64, -1.47] | 0.09 [-0.23, 0.47] |
| Ex2 | a | 102.6 [101.6, 103.8] | 102.9 [101.5, 104.4] | 0.3 [-0.9, 1.5] |
|  | b | -0.031 [-0.044, -0.020] | -0.030 [-0.043, -0.017] | 0.002 [-0.002, 0.006] |
| Ex3 | a | 76.3 [65.1, 95.3] | 63.4 [55.0, 75.2] | -12.7 [-25.1, -4.5] |
|  | b | -0.045 [-0.068, -0.022] | -0.054 [-0.080, -0.032] | -0.010 [-0.021, -0.001] |
|  | c | 27.3 [7.2, 38.1] | 40.7 [27.9, 48.8] | 13.7 [5.2, 25.9] |
| Crit | L' | 3638.8 [2062.5, 6422.1] | 4583.0 [2637.9, 7271.6] | 613.7 [287.6, 1129.9] |
|  | k | -32.0 [-47.5, -20.5] | -36.5 [-52.0, -24.3] | -4.1 [-6.6, -2.0] |
|  | CL | -17.4 [-43.4, 3.6] | -23.5 [-48.5, -2.5] | -3.3 [-8.8, 0.0] |

Posterior predictive distributions are summarized using the Maximum a Posteriori estimate and 90% Highest Density Interval.

Crit, critical load model; Ex2, exponential model (2 parameters); Ex3, exponential model (3 parameters); Lin, linear model; $\Delta x_i$, change effect between T1 and T2.

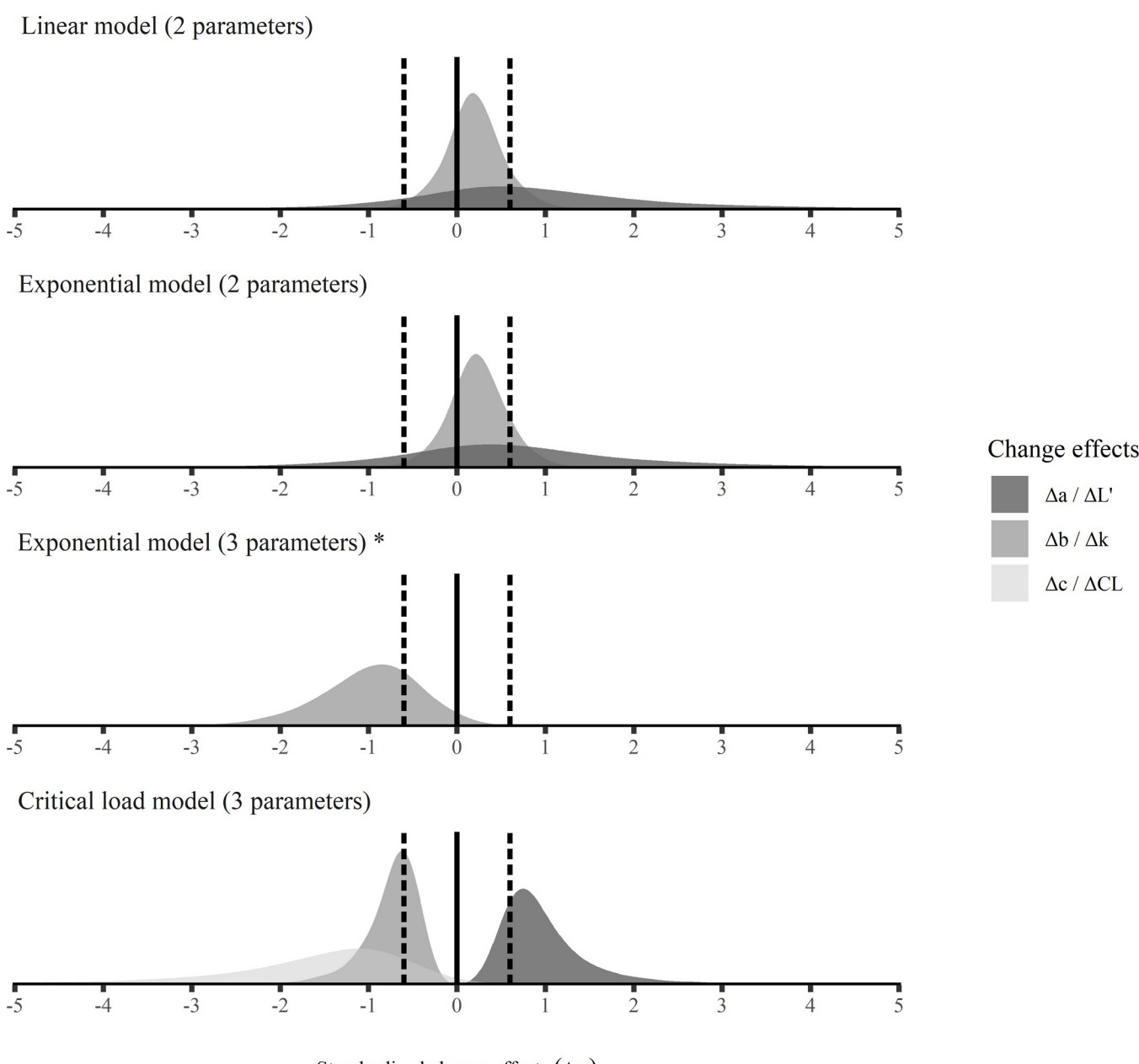

**Fig 3. Posterior predictive distributions for standardized subject-level change effects (smoothed illustration).** Dashed black lines, threshold for acceptable differences set to [-0.6, 0.6] indicating small or trivial changes; *, change effects *Δa* and *Δc* of the exponential 3-parameters model are not visibly displayed due to very large scales.

performed at 70%, 80% and 90% 1-RM. However, a slight shift of SEM posterior distributions to lower values could be observed for RTF at higher relative loads. In particular, the difference of SEM between RTF at 70% and 90% 1-RM could have exceeded the predefined threshold for systematic differences at a larger sample size. Interestingly, the ICC showed an opposing non-systematic shift of posterior distributions, with lower relative loads resulting in slightly larger ICC values. These seemingly contradictory trends arising from absolute and relative consistency might be related to the computation of the respective statistics: in the present study, the

**Table 3. Summary of posterior predictive distributions of relative and standardized change effects.**

| Model | Change effect | Relative magnitude (%) * | Standardized magnitude ** | p ($\Delta x_i \in$ [-0.6, 0.6] | data) ** |
|---|---|---|---|---|
| Lin | $\Delta a$ | 0.3 [-0.8, 1.6] | 0.38 [-4.46, 8.98] | 27.6% |
| | $\Delta b$ | 4.8 [-8.9, 17.2] | 0.17 [-0.38, 0.78] | 86.9% |
| Ex2 | $\Delta a$ | 0.4 [-1, 1.4] | 0.13 [-4.43, 9.35] | 29.5% |
| | $\Delta b$ | 4.6 [-7.9, 18.1] | 0.22 [-0.36, 0.8] | 84.8% |
| Ex3 | $\Delta a$ | -19.1 [-28.5, -7.8] | -15.26 [-184.85, 0.71] | 0.2% |
| | $\Delta b$ | -20.4 [-48.5, -0.7] | -0.83 [-2, -0.04] | 26.1% |
| | $\Delta c$ | 42.1 [0.4, 217.4] | 18.34 [0.05, 232.88] | 0.1% |
| Crit | $\Delta L'$ | 14.6 [6.7, 29.6] | 0.76 [0.24, 1.83] | 18.8% |
| | $\Delta k$ | -12.1 [-22.1, -6] | -0.63 [-1.26, -0.24] | 36.2% |
| | $\Delta CL$ | -10.9 [-129.6, 7.8] | -1.2 [-6.4, 0.52] | 11.5% |

Posterior predictive distributions are summarized using the Maximum a Posteriori estimate and 90% Highest Density Interval.

*, change effects are expressed relative to the group-level mean of the associated model parameter at T1.

**, change effects are standardized to the group-level standard deviation of the associated model parameter at T1. Crit, critical load model; Ex2, exponential model (2 parameters); Ex3, exponential model (3 parameters); Lin, linear model; p ($\Delta x_i \in$ [-0.6, 0.6] | data), probability of the standardized change effect falling within the threshold for acceptable differences given the data.

ICC was calculated as the proportion of total variance attributed to the variance among subjects. Therefore, it tends to be smaller when between-subject variance is low and SEM is large. Indeed, our data suggest a higher between-subject variance of the RTF at 70% 1-RM compared to 90% 1-RM. A similar trend for heteroscedasticity in the relationship between relative load and RTF across individuals (i.e. a mean-variance "tradeoff") has been reported on numerous occasions [3,11,12,14,18,19,37,38]. This phenomenon could be the result of normalizing the load to the 1-RM, which homogenizes the upper end of the load spectrum. However, it could also be partially explained by inter-individual differences in the strength-endurance relationship.

Conforming trends for the reliability of the RTF performed at given relative loads can be observed from other sources. For example, Anders and colleagues reported an ICC of 0.90 (95% CI: [0.58, 0.97]) for RTF completed at 70% 1-RM in the bench press [4], indicating a similar magnitude compared to the present study (ICC [90% HDI] = 0.86 [0.71, 0.93]). While the reported SEM of 0.68 repetitions was noticeably lower compared to the present study, the authors also described a lower between-subject standard deviation of ±1.5 repetitions. Similarly, Pereira and colleagues reported an ICC of 0.90 for the RTF achieved at 75% 1-RM in the bench press, when performing repetitions at a joint velocity of 100°/s. While no information on subject heterogeneity was provided, the authors also reported an ICC of 0.70 when the exercise was completed at a joint velocity of 25°/s. It could be hypothesized that the reduced movement cadence might have negatively affected the number of repetitions performed [14,24], possibly due to an increased duration of the concentric phase of each repetition and associated increases in metabolic demand [39]. Hence, a reduced movement cadence at lower loads could result in a distribution of RTF that is similar to the RTF at higher loads when repetitions are performed at maximal voluntary velocity, as was the case in the present study.

Other studies investigated the reproducibility of RTF in absolute loads. For example, Mann et al. analyzed the test-retest reliability of NCAA Division I football players in the NFL-225 test, which is a repetition maximum test using a fixed load of 225 lbs or 102.3 kg in the bench press exercise [40]. The authors reported an ICC of 0.98 to 0.99 and a typical error of 1.0 to 1.3 repetitions across three trials, the typical error corresponding to what has been calculated as

SEM in the present study. While it is difficult to evaluate at what percentage of the 1-RM each participant performed the NFL-225 test in the absence of a 1-RM test, the authors estimated it to be around 67.9% 1-RM for athletes with a body mass below 100.5 kg and around 44.6% 1-RM for heavier athletes. Therefore, the majority of participants performed the NFL-225 test at lower relative loads compared to the present study. Given this fact, the reports of Mann et al. [40] correspond well to the results of the present study (SEM for RTF at 70% 1-RM [90% HDI] = 1.1 [0.8, 1.4] repetitions), especially when considering the large between-subject variance reported by the authors, which may have contributed to the large ICC, as discussed before. Finally, Rose and Ball analyzed the reliability of the RTF that could be achieved against 15.9 kg and 20.4 kg, reporting an ICC of 0.97 in both cases [36]. In their sample of 21 moderately trained women the two tested loads corresponded roughly to a mean relative load of 42% and 54% 1-RM, which supports the hypothesis of RTF tests showing higher relative consistency at low loads.

A systematic increase in the 1-RM between test and retest has previously been described on numerous occasions for various exercises [41]. Interestingly, Ribeiro and colleagues reported that this time effect did not interact significantly with participants' experience in resistance training [42]. While the magnitude of the systematic change ($\Delta t$ [90% HDI] = 1.9 kg [1.0, 2.7]) could be considered trivial in the present study, given the smallest load increment was 2.5 kg, previous research suggested that the effect may occur over the course of multiple consecutive retest trials as a result of practicing the test [42–44]. Similarly, the time effect of RTF performed at 90%, 80% and 70%-1RM showed a high probability for being less than 1 repetition. Despite the RTF at 80% and 70% 1-RM indicating a systematic difference between T1 and T2, the magnitude of this effect is likely trivial.

To the best of our knowledge, this is the first study to evaluate and compare the reproducibility of different strength-endurance models with respect to individual trends. Not all of the investigated models resulted in robust parameter estimates over time. Most notably, the 3-parameters exponential model and the critical load model exhibited systematic changes for all parameters. These findings suggest that naturally occurring variability in strength performance likely causes parameter estimates to systematically change, even over short periods, and that the magnitude of these changes is unacceptably high in relation to the respective parameter's group-level standard deviation. Therefore, the two models may not provide sufficient reproducibility for application in the practical field. In comparison, the linear model and the 2-parameters exponential model both resulted in a high probability for $\Delta b$ (i.e., the change in slope and curvature parameters, respectively) to fall within the threshold for acceptable differences, although the effects were not clear at the selected credibility level. No clear change effect could be identified for the intercept parameter $a$ in both cases due to low between-subject variability. However, findings suggest a negligible relative magnitude for $\Delta a$ in both models (Table 3). Therefore, both the linear model and 2-parameters exponential model yield the most robust parameter estimates across test-retest trials among the investigated models. To decide which of the two models to apply in a practical setting, practitioners should also consider statistical qualities other than the robustness of models. For example, both the model fit und predictive validity can be considered essential characteristics of a valuable strength-endurance profile. While previous research provided some evidence that the relationship may be considered approximately linear at high loads [3,10–12], it has been suggested that the relationship actually follows a curvilinear trend when considering the full spectrum of loads [11,13,14]. Therefore, practitioners might want to resort to applying the 2-parameters exponential regression rather than the linear regression to model strength-endurance profiles, as research has not proposed any explicit disadvantages reasoning against its use.

Based on the findings of the present study, a freely available web application was developed using the R package *shiny* (version 1.7.1). The application provides practitioners with a user-friendly interface to enter data from repetition maximum tests and offers different algorithms to compute the individual and exercise-specific strength-endurance profile. Upon computation, it offers a graphical display of the profile, a model equation and an adjusted $R^2$ estimate to evaluate model fit. Furthermore, it produces an individual repetition-maximum table based on the estimated model parameters that predicts loads for a wider spectrum of RTF. A link to the web application is provided at the end of this article.

It should be pointed out that the order of repetition maximum tests was not randomized in the present study. Hence, a possible systematic effect of the earlier sets performed to momentary failure on subsequent sets and, thus, the presence of systematic bias in the RTF performed cannot be excluded. Future research should strive to compare different test protocols and identify a valid, yet practically applicable approach to acquiring the necessary data for model computation. However, the results of the present study may help practitioners understand the consistency of strength performance under standardized conditions and can assist with the selection of a reliable statistical model to calculate individual strength-endurance profiles.

## Conclusions and practical applications

In conclusion, both the 1-RM and RTF at 90%, 80% and 70% 1-RM showed good reproducibility over test-retest trials in the bench press exercise for trained subjects. When modeling the relationship between load and RTF using a multilevel structure, the linear regression and 2-parameters exponential regression provide more stable parameter estimates than the 3-parameters exponential regression or critical load model.

To calculate a strength-endurance profile for a given individual and specific exercise, it is recommended to acquire the maximum number of repetitions that can be performed to momentary failure against three different loads. While the loads should be chosen according to a range of interest, practitioners should expect to experience higher absolute day-to-day variability of RTF at lower loads. For loads in the range of 70% - 100% 1-RM, a linear regression or a 2-parameters exponential regression should be applied to reliably model the relationship between tested loads and the number of achieved repetitions. To derive a robust strength-endurance profile, practitioners can access a free-to-use web application using the following link: https://strength-and-conditioning-toolbox.shinyapps.io/Strength-Endurance_Profile/.

## Supporting information

**S1 Appendix. Modeling details.** This file contains detailed information on priors and models. (PDF)

**S1 Table. Raw data.** This file contains the data used for the statistical analyses. (XLSX)

## Acknowledgments

The authors would like to thank all participants for contributing to the realization of the present study. The authors further want to express their gratitude to the Centre for Sport Science and University Sports, University of Vienna for providing the equipment and facilities.

## Author Contributions

**Conceptualization:** Benedikt Mitter, Harald Tschan.

**Data curation:** Benedikt Mitter.

**Formal analysis:** Benedikt Mitter.

**Investigation:** Benedikt Mitter.

**Methodology:** Benedikt Mitter.

**Project administration:** Benedikt Mitter, Harald Tschan.

**Resources:** Benedikt Mitter, Robert Csapo, Harald Tschan.

**Software:** Benedikt Mitter.

**Supervision:** Harald Tschan.

**Visualization:** Benedikt Mitter, Pascal Bauer.

**Writing – original draft:** Benedikt Mitter, Robert Csapo, Pascal Bauer, Harald Tschan.

**Writing – review & editing:** Robert Csapo, Pascal Bauer, Harald Tschan.

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
