## [Decision Letter · Decision Letter 0]

21 Feb 2022

PONE-D-22-01272Reproducibility of strength performance and strength-endurance profiles: a test-retest studyPLOS ONE

Dear Dr. Mitter,

Thank you for submitting your manuscript to PLOS ONE. After careful consideration, we feel that it has merit but does not fully meet PLOS ONE’s publication criteria as it currently stands. Therefore, we invite you to submit a revised version of the manuscript that addresses the points raised during the review process.

ACADEMIC EDITOR Dear authors. Thank you for submitting your MS to Plos One. As you will see below, both reviewers found your study interesting and rigorously conducted. They also raised some important concerns which must be adressed during the revision process. Pay particular attention at improving the discussion section according to reviewers' suggestions. Best wishesMathieu Gruet==============================

We look forward to receiving your revised manuscript.

Kind regards,

Mathieu Gruet, Ph.D

Academic Editor

PLOS ONE

Journal Requirements:

Reviewers' comments:

Reviewer's Responses to Questions

**Comments to the Author**

1. Is the manuscript technically sound, and do the data support the conclusions?

Reviewer #1: Yes

Reviewer #2: Yes

2. Has the statistical analysis been performed appropriately and rigorously? 

Reviewer #1: Yes

Reviewer #2: Yes

3. Have the authors made all data underlying the findings in their manuscript fully available?

Reviewer #1: Yes

Reviewer #2: Yes

4. Is the manuscript presented in an intelligible fashion and written in standard English?

Reviewer #1: No

Reviewer #2: Yes

5. Review Comments to the Author

Reviewer #1: The present study was designed to evaluate the test-retest consistency of repetition maximum tests at standardized relative loads and determine the robustness of strength-endurance profiles across test-retest trials. The topic is of interest. The introduction & results are well-written parts. As specified in my comments, I think that the methods parts needs some clarification, especially to better understand the experimental design. Finally, the discussion appears as the weakest point of this manuscript, and needs to be amended before the manuscript could be accepted for publication. For the form, some typographical or grammatical errors need to be addressed. Please see my main comments below.

Introduction:

Line 30: please give more explanation for absolute vs relative loads. I guess that “relative” refers to RM; but please justify it since other subjective methods exist (e.g. RIR).

Line 35-36: I understand your rational, but some methods that predict the load associated with a certain repetition maximum already exist. It could be good to strengthen this point.

Line 37: what is the interest to lead a repetition set to exhaustion?

Line 44: while the prediction become lower when the number of repetitions is higher?

Methods:

Line 75: Something is wrong in the formulation of this sentence; i.e. “twenty-four resistance-trained men (n=15)” is confusing. Please amend this sentence. Further, I am not sure that the BP 1-RM/body mass needs to appear here.

Line 92: is “work capacity” the best formulation to be used? Should you not refer to “endurance” here (as mainly performed in the introduction)?

Lines 93-94: did you control (objectively of subjectively) for muscle damage potentially induced by the 1-RM test before the beginning of the repetition maximum tests?

Lines 95-97: this is not clear. The determination of the 1-RM was not the first visit? You talk of 2 visits for T1 and T2; just after you say that participants visited the laboratory on four occasions. I think that this paragraph is important, and as it stands it is not clear (maybe a figure summarizing the experimental design could help).

Line 119-120: what does this mean to “subjectively estimate the 1RM”?

Line 132: what was the duration of the break? This is an important feature here to make sure that no fatigue was present before each rep max tests.

Lines 131-142: what about the eccentric phase during the rep max test? There is no information on that? Was it passive or active? Were specific instructions given for this eccentric phase?

Line 141: why did you chose 22min for the resting duration?

Discussion:

The main results of your study (rather than only an objectives reminder) should appear in the first paragraph. The results section could be complicated for some readers that are not used to mathematical models, thus summarizing the main results at the beginning of the discussion could be helpful. Further, the last sentence (line 254-256) would fit better at the end of the discussion for your perspectives.

Line 263-264: I am not sure that a difference of 0.5 rep between 70% and 90 & 80% RM test could suggest (even if this is only a suggestion) that the test-retest agreement is better at higher relative loads.

Line 268: you cannot put “:” and start the following sentence with a majuscule. This appears many times in your manuscript. Please correct.

Lines 270-272: do you have objective explanation for this result?

Line 274: it could be helpful to remind the ICC obtained in your study to confront it with the one of Anders et al.

Line 282: why a decrease in movement cadence can affect the number of repetitions?

In general, the discussion lacks depth. We do not clearly identify the main results of the study and the application that these results could have. Since the results part is a bit difficult to understand (although the statistical analysis are relevant), a clear and detailed discussion (with objective clues) is needed. I am further surprised by the small number of references that are used in the discussion. The development of a web application is an interested feature of this article, and should be more highlighted in my opining rather that only cited in the last sentence of the conclusion.

Reviewer #2: I’d like to congratulate the authors on a simple yet elegant, and rigorously conducted, study. I actually have very little to suggest here and think that it could largely be published as is. The app is very nicely put together too. I just make a few comments below which the authors might wish to consider.

Many thanks

James Steele

Comments:

“Failure” – you use the phrase “volitional failure” and do not provide a definition for this. You might wish to consider the following article from our group that discusses definitions of terms in relation to this - https://pubmed.ncbi.nlm.nih.gov/28044366/

In a supplementary analysis for a recent meta-analysis from our group (https://sportrxiv.org/index.php/server/preprint/view/109/version/120), we collated data from some studies (https://osf.io/td26u/) reporting group level results for repetitions performed to failure at different relative loads. We did explore group level strength-endurance profiles (though in order to compare self-selected repetitions numbers to what could be performed; https://osf.io/xqz9a/). Anyway, I just thought it might be of interest considering this current study. We only fit a simple linear model to it, but it would be interesting to see how well the other models you describe might fit.

I appreciate the reasoning for not randomising the loads, though think it might be worth mentioning that this is a possible limitation that could in and of itself introduce some degree of systematic bias. Perhaps just mention it in the discussion.

I might also add to the statistical analysis when describing the models for strength-endurance profiles that the random effects for participants included both intercepts and slopes. It is clear from the equations, but not all are mathematically inclined and so explicitly mentioning this in the text might be worthwhile.

The small systematic increase in 1RM, and perhaps RTFs, might be explained by the test practice effect as Jeremy Loenneke’s group have discussed (e.g., https://pubmed.ncbi.nlm.nih.gov/27875635/, https://pubmed.ncbi.nlm.nih.gov/28463902/)

I think it would be worthwhile to include, similarly to the 1RM/RTF table, a table showing the parameter estimates from each model for T1 and T2 in addition to the change parameter estimated. It would be nice for example to compare to estimates from other studies (I appreciate the data are available so a reader could do this themselves if they wanted too though).

Lastly, you mention the mean-variance relationship in the discussion. I just thought it worth highlighting that this is very much apparent for repetitions performed, particularly for their log transformation (see meta-analytic estimate from the supplementary data in our meta-analysis mentioned: https://osf.io/fznhu?show=view&view_only=).

6. PLOS authors have the option to publish the peer review history of their article (what does this mean?). If published, this will include your full peer review and any attached files.

Reviewer #1: **Yes: **Robin Souron

Reviewer #2: **Yes: **James Steele

---

## [Author Response · Author response to Decision Letter 0]

30 Mar 2022

Overall response: We would like to express our gratitude to both reviewers for providing constructive feedback on our manuscript and giving valuable suggestions to improve the scientific quality of the article. We believe that with the help of the editor and the reviewers, we managed to eradicate any remaining ambiguities, improve the methodological transparency and provide readers a more profound discussion of our results. 

Response to reviewer 1

R1: The present study was designed to evaluate the test-retest consistency of repetition maximum tests at standardized relative loads and determine the robustness of strength-endurance profiles across test-retest trials. The topic is of interest. The introduction & results are well-written parts. As specified in my comments, I think that the methods parts needs some clarification, especially to better understand the experimental design. Finally, the discussion appears as the weakest point of this manuscript, and needs to be amended before the manuscript could be accepted for publication. For the form, some typographical or grammatical errors need to be addressed. Please see my main comments below.

Response: Thank you for taking the time to thoroughly review our manuscript, pointing out missing details and suggesting improvements to enhance both the practical and scientific value of the article. We hope the adjustments made are to the satisfaction of the reviewer.

Introduction:

R1.C1: Line 30: please give more explanation for absolute vs relative loads. I guess that “relative” refers to RM; but please justify it since other subjective methods exist (e.g. RIR).

Response: The authors would like to thank the reviewer for pointing out an ambiguous expression. We provided readers with a description of how absolute and relative loads should be understood in the context of the present study. 

Before: […] an exercise being performed to volitional failure at either a fixed absolute or relative load.

After: Line 30-31: […] an exercise being performed to volitional failure at either a fixed absolute load, expressed in a unit of mass like kg or lbs, or a fixed relative load that has been normalized to the exercise-specific one-repetition maximum (1-RM).

R1.C2: Line 35-36: I understand your rationale, but some methods that predict the load associated with a certain repetition maximum already exist. It could be good to strengthen this point.

Response: Thank you for the pertinent suggestion. Indeed, research on the relationship between load and repetition maximum (or RTF) has a long history in exercise science. In the second paragraph of the introduction (Line 47-59), readers are provided with a short overview of published bivariate models and the limitations of a modeling approach that generalizes parameters across individuals. 

We rephrased the statement addressed by the reviewer to avoid its misinterpretation as a claim for exclusivity (i.e., the individual “strength-endurance profile” being the only method to predict loads from a repetition maximum, which would not be correct). 

Before: […] by studying the relationship between load and RTF (i.e., the individual “strength-endurance profile”) which would also enable practitioners to predict the load associated with a certain repetition maximum.

After: Line 37-38: […] by studying the relationship between load and RTF (i.e., the individual “strength-endurance profile”). Additionally, knowledge of the mathematical relationship between the two variables could be used by practitioners to predict the load associated with a certain repetition maximum.

R1.C3: Line 37: what is the interest to lead a repetition set to exhaustion?

Response: Thank you for requesting clarification. We believe the reviewer may have misinterpreted out statement, eventually due to an inappropriate choice of terminology (i.e., “exhaustion”). We considered “intensity of effort” might be more suitable, since a recent article explicitly proposed its definition [1]. 

The sentence addressed by the reviewer was referring to the idea of controlling intensity of effort on a submaximal level, therefore not executing a set to momentary failure. To avoid misinterpretation by readers, we included a comparison to autoregulatory approaches of controlling intensity of effort.

Before: This may be of particular interest for individuals seeking to control within-set exhaustion by prescribing […]. 

After: Line 39-40: This may be of particular interest for individuals seeking to control intensity of effort within a set [1] by prescribing […].

Added sentences: Line 41-46: While other methods have been proposed to evaluate or control intensity of effort based on perceived effort or movement velocity [2], an approach using strength-endurance profiles might overcome certain limitations of these methods. Such limitations include inappropriate anchoring of perception [1], inaccurate subjective estimates of repetitions in reserve at lower intensity of effort [3] and dependency on technology to provide reliable feedback on movement velocity [4]. 

R1.C4: Line 44: while the prediction become lower when the number of repetitions is higher?

Response: The authors would like to thank the reviewer for his comment. If we understand correctly, the reviewer is addressing the influence of the number of repetitions performed in the RM test on prediction bias. Some of the referenced studies suggest a trend of most predictive equations overestimating the 1-RM with decreasing load and an increasing number of repetitions performed [5, 6]. Others reported a trend of underestimating the 1-RM at lower repetition numbers (<10) in the RM test [7, 8] or only showed systematic prediction bias for specific equations [9, 10]. 

In the opinion of the authors, an extensive discussion of validation studies would not be beneficial to the introduction of the present manuscript, as published research does not indicate a homogenous trend for prediction bias that applies to all predictive equations, exercises and sample characteristics. We believe that prediction bias of respective equations is a multifactorial phenomenon that has yet to be investigated comprehensively. However, it was not the objective of the present study, and the project the study is embedded within, to analyze the shortcoming of these equations, but to investigate an alternative approach as a potential solution to the problem. Therefore, we decided not to go into further detail on the addressed validation studies.

Methods:

R1.C5: Line 75: Something is wrong in the formulation of this sentence; i.e. “twenty-four resistance-trained men (n=15)” is confusing. Please amend this sentence. Further, I am not sure that the BP 1-RM/body mass needs to appear here.

Response: Thank you for pointing out a confusing statement. We amended the sentence as suggested by the reviewer. Concerning the bench press 1-RM/body mass statistics, we believe it may be considered a valuable piece of information to verify that participants did indeed have substantial training experience. We also believe that the tested 1-RM is to be preferred over self-reported indicators of training experience in terms of validity and precision. Since we also defined inclusion criteria for 1-RM/body mass in the “Subjects” section, we thought it might be more appropriate to report 1-RM/body mass statistics here, rather than at the beginning of the results section. 

Before: Twenty-four resistance-trained men (n = 15, age = 27.2 ± 3.3 yrs, body mass = 85.4 ± 7.9 kg, bench press 1-RM/body mass = 1.33 ± 0.11 kg•kg-1) and women (n = 9, age = 27.7 ± 5.2 y, body mass = 63.6 ± 3.3 kg, bench press 1-RM/body mass = 0.96 ± 0.17 kg•kg-1) volunteered […]

After: Line 83-84: Fifteen resistance-trained men (age = 27.2 ± 3.3 yrs, body mass = 85.4 ± 7.9 kg, bench press 1-RM/body mass = 1.33 ± 0.11 kg•kg-1) and nine resistance-trained women (age = 27.7 ± 5.2 y, body mass = 63.6 ± 3.3 kg, bench press 1-RM/body mass = 0.96 ± 0.17 kg•kg-1) volunteered […]

R1.C6: Line 92: is “work capacity” the best formulation to be used? Should you not refer to “endurance” here (as mainly performed in the introduction)?

Response: Thank you for this valuable suggestion. Indeed, we only used the term “work” at the very beginning of the introduction and it might confuse readers to switch back and forth. We changed the expression using “strength-endurance”. 

Before: A test-retest design was used to determine the participants’ maximum strength and work capacity at high loads […]

After: Line 97: A test-retest design was used to determine the participants’ maximum strength and strength-endurance at high loads […]

Before: Work capacity was assessed using […]

After: Line 99: Strength-endurance was assessed using […]

R1.C7: Lines 93-94: did you control (objectively of subjectively) for muscle damage potentially induced by the 1-RM test before the beginning of the repetition maximum tests?

Response: The authors would like to thank the reviewer for this pertinent comment. No biological markers of muscle damage were collected over the course of the present study, as we wanted to omit any invasive measurements. Furthermore, no subjective markers of muscle damage were assessed to quantify the magnitude of muscle damage, as we considered it difficult to apply a valid anchoring process (i.e. “calibrating” participants’ perception of a maximum score) without a priori matching it to biological markers of muscle damage. 

We did assess barbell velocity using a linear position transducer throughout every set of the test protocol. Recently, a study suggested that velocity-based estimates of the 1-RM, despite not providing good predictions of the actual 1-RM, might reflect training-induced fatigue to some degree [11]. When following a similar approach to what the authors called a MVT-based 1-RM estimate [11], using the highest mean velocity of the 3 initial warm-up sets (approx.. 25%, 50% and 75% 1-RM) of the 1-RM test (session 1) and repetition maximum tests (session 2) for model computation, the resulting MVT-based 1-RM estimates did not indicate a clear decrease between session 1 and session 2 (mean difference ± SD = -1.6 ± 4.3 kg). However, we acknowledge that this indirect approach does not necessarily prove the absence of muscle damage during repetition maximum tests 48-72h after the 1-RM test. Therefore, we decided not to include it in the present manuscript, as we believe it would not contribute to the main objective of the article. 

We would argue that even if muscle damage was present over the course of the 4 sessions of this study, both the test and retest (T1 and T2) were still executed with a standardized test protocol including standardized timing between the 1-RM test and the repetition maximum test at T1 and T2. Therefore, we believe the robustness of models would not have been systematically affected by eventual muscle damage. 

R1.C8: Lines 95-97: this is not clear. The determination of the 1-RM was not the first visit? You talk of 2 visits for T1 and T2; just after you say that participants visited the laboratory on four occasions. I think that this paragraph is important, and as it stands it is not clear (maybe a figure summarizing the experimental design could help).

Response: Thank you for pointing out an unclear description of the experimental protocol. We added a figure describing the experimental setup and referenced it in the addressed section. The numbering of the subsequent figures was adapted accordingly.

Added sentence: Line 109-110: Fig 1. Experimental design. 1-RM, one-repetition maximum; RTF, repetitions performed to momentary failure.

Before: […] on two occasions (T1 & T2) separated by one week.

After: Line 98: […] on two occasions (T1 & T2) separated by one week (Figure 1).

Before: Fig.1 / Figure 1

After: Line 238 / Line 227: Fig.2 / Figure 2

Before: Fig.2 / Figure 2

After: Line 267 / Line 248: Fig.3 / Figure 3

R1.C9: Line 119-120: what does this mean to “subjectively estimate the 1RM”?

Response: The authors would like to thank the reviewer for suggesting further clarification and apologize for the ambiguous expression. We rephrased the sentence accordingly.

Before: […] participants were requested to subjectively estimate their 1-RM.

After: Line 130: […] participants were requested to estimate their 1-RM based on self-evaluation of their recent training performance.

R1.C10: Line 132: what was the duration of the break? This is an important feature here to make sure that no fatigue was present before each rep max tests.

Response: Thank you for requesting further details. The time in between rep max tests (~22 min) is mentioned at the bottom of the paragraph and was added to Figure 1. The addressed paragraph was revised, since the term “break” might suggest the application of purely passive rest. However, as described in the paragraph, the time in between rep max tests was also used to actively sustain warm-up effects, therefore providing the same warm-up routine prior to each rep max test.

Investigating the presence of fatigue induced by a single visit protocol was beyond the scope of the present study. However, we would like to inform the reviewer that our laboratory recently started data acquisition for a study on that topic, comparing a single-visit protocol to a multiple-visit protocol in a crossover design using stratified randomization. 

Before: […] in the form of a single-visit protocol with prolonged breaks in between.

After: […] in the form of a single-visit protocol.

Before: In order to sustain the warm-up effect during these prolonged breaks, participants underwent the same general warm-up procedure used for the 1-RM test prior to each set to failure. 

After: Line 144-146: In order to provide extended time for recovery in between repetition maximum tests, yet sustain warm-up effects during these periods, participants underwent the same general warm-up procedure that was used for the 1-RM test prior to each set to failure.

R1.C11: Lines 131-142: what about the eccentric phase during the rep max test? There is no information on that? Was it passive or active? Were specific instructions given for this eccentric phase?

Response: Thank you for suggesting further improvements to promote the reproducibility of our experimental protocol. The requested details were added to the paragraph accordingly. The authors would like to emphasize that the use of a fixed movement cadence was abandoned on purpose for the rep max tests, as a recent review explicitly suggested this aspect for future investigations [12]. 

Added sentences: Line 153-158: Participants were instructed to lower the barbell in a controlled fashion on each repetition, albeit not being prescribed a fixed movement cadence. Similar to the 1-RM test, participants had to await the verbal command of the staff member before initiating the concentric phase of a repetition, in order to avoid any rebound from the safety pins. The concentric phase of each repetition had to be performed at maximum intended velocity. 

R1.C12: Line 141: why did you chose 22min for the resting duration?

Response: Thank you for requesting clarification. The time interval in between rep max tests was the result of the described standardized warm-up procedure and the 5 min of rest provided before and immediately after each rep max test. We acknowledge that the protocol resulting in the addressed time interval was, to some extent, an arbitrary choice by the authors in an attempt to balance recovery and duration for the session, while maintaining positive warm-up effects across all tests. However, investigating the effects of different rest intervals during a single visit assessment was beyond the scope of the present study. We would further argue that the length of the rest interval does not necessarily bias any estimate for reproducibility of performance or model robustness, as long as the protocols for T1 and T2 are identical (as is the case in the present study). 

The addressed section was adapted to explain readers where the 22 min resulted from. 

Before: Due to this methodological structure, the repetition maximum tests were separated by approximately 22 min each.

After: Line 150-151: Due to this methodological structure (i.e., the standardized warm-up; the standardized passive rest before and after each set to failure), the repetition maximum tests were separated by approximately 22 min each.

Discussion:

R1.C13: The main results of your study (rather than only an objectives reminder) should appear in the first paragraph. The results section could be complicated for some readers that are not used to mathematical models, thus summarizing the main results at the beginning of the discussion could be helpful. Further, the last sentence (line 254-256) would fit better at the end of the discussion for your perspectives.

Response: We would like to thank the reviewer for suggesting structural improvements. Changes were made accordingly. 

Added sentences: Line 277-280: Test-retest analysis of performance indicated very good reproducibility of the 1-RM and the RTF at high relative loads in the bench press exercise. The linear regression and the 2-parameters exponential regression yielded the most robust parameter estimates across the investigated models of the strength-endurance relationship. 

Deleted: Our results may help practitioners understand the consistency of strength performance and assist with the selection of a reliable statistical model to calculate individual strength-endurance profiles. 

Added sentence: Line 386-389: However, the results of the present study may help practitioners understand the consistency of strength performance under standardized conditions and can assist with the selection of a reliable statistical model to calculate individual strength-endurance profiles.

R1.C14: Line 263-264: I am not sure that a difference of 0.5 rep between 70% and 90 & 80% RM test could suggest (even if this is only a suggestion) that the test-retest agreement is better at higher relative loads.

Response: The authors would like to thank the reviewer for pointing this out. Indeed, the difference in SEM could not be deemed clear at the 90% credibility level and we agree that the discussion of our results should avoid suggestive statements that are not supported by our predefined thresholds for qualitative interpretation. However, given that the width of posterior distributions and, hence, the certainty about differences in effects is typically affected by sample size, we believe that any effect that missed the threshold by a small percentage of probability mass should still be pointed out. 

Before: […] whereas the absolute agreement was better at higher loads […]

After: Line 14: […] whereas the absolute agreement was slightly better at higher loads […]

Deleted sentences: While these results would suggest that test-retest agreement for the RTF is better at higher relative loads compared to lower relative loads, the relative consistency of RTF performance was found to be slightly worse at higher relative loads compared to lower relative loads, although the difference in magnitude of the ICC could not be deemed systematic at the 90% credibility level. 

Added sentences: Line 287-293: Posterior distribution analysis revealed no systematic differences of SEM between RTF performed at 70%, 80% and 90% 1-RM. However, a slight shift of SEM posterior distributions to lower values could be observed for RTF at higher relative loads. In particular, the difference of SEM between RTF at 70% and 90% 1-RM could have exceeded the predefined threshold for systematic differences at a larger sample size. Interestingly, the ICC showed an opposing non-systematic shift of posterior distributions, with lower relative loads resulting in slightly larger ICC values. 

Before: The seemingly contradictory findings arising from absolute and relative consistency can, however, be explained mathematically to some degree

After: Line 293-294: These seemingly contradictory trends arising from absolute and relative consistency might be related to the computation of the respective statistics […]

R1.C15: Line 268: you cannot put “:” and start the following sentence with a majuscule. This appears many times in your manuscript. Please correct.

Response: Thank you for pointing this out. Changes were made accordingly.

Before: […] for the bench press: Each subject […]

After: Line 121: […] for the bench press: each subject […]

Before: […] two objectives: First, we evaluated […]

After: Line 273: […] two objectives: first, we evaluated […]

Before: […] to some degree: In the present study […]

After: Line 294: […] to some degree: in the present study […]

R1.C16: Lines 270-272: do you have objective explanation for this result?

Response: Thank you for proposing further details on the heteroscedasticity of the strength-endurance relationship. We adapted this section and provided readers with a possible explanation. Adaptations were also made as part of our response to the other reviewer’s comment #7. 

Before: Indeed, our data suggest a higher between-subject standard deviation of the RTF at 70% 1-RM compared to 90% 1-RM.

After: Line 297-303: Indeed, our data suggest a higher between-subject variance of the RTF at 70% 1-RM compared to 90% 1-RM. A similar trend for heteroscedasticity in the relationship between relative load and RTF across individuals (i.e. a mean-variance “tradeoff”) has been reported on numerous occasions [5, 9, 13–18]. This phenomenon could be the result of normalizing the load to the 1-RM, which homogenizes the upper end of the load spectrum. However, it could also be partially explained by inter-individual differences in the strength-endurance relationship.

R1.C17: Line 274: it could be helpful to remind the ICC obtained in your study to confront it with the one of Anders et al.

Response: Thank you for this pertinent comment. Readers were provided the associated ICC value of the present study, as suggested by the reviewer.

Before: […] Anders and colleagues reported an ICC of 0.90 (95% CI: [0.58, 0.97]) for RTF completed at 70% 1-RM in the bench press [19]. 

After: Line 306-307: […] Anders and colleagues reported an ICC of 0.90 (95% CI: [0.58, 0.97]) for RTF completed at 70% 1-RM in the bench press [19], indicating a similar magnitude compared to the present study (ICC [90% HDI] = 0.86 [0.71, 0.93]).

R1.C18: Line 282: why a decrease in movement cadence can affect the number of repetitions?

Response: Thank you for requesting clarification. We expanded upon our hypothetical explanation based on past research, as suggested by the reviewer. 

Before: It could be hypothesized that the reduced movement cadence might negatively affect the number of repetitions that can be achieved at the respective load, therefore resulting in a distribution similar to the RTF at higher loads when repetitions are performed at maximal voluntary velocity, as was the case in the present study.

After: Line 314-317: It could be hypothesized that the reduced movement cadence might have negatively affected the number of repetitions performed [16, 20], possibly due to an increased duration of the concentric phase of each repetition and associated increases in metabolic demand [21]. Hence, a reduced movement cadence at lower loads could result in a distribution of RTF that is similar to the RTF at higher loads when repetitions are performed at maximal voluntary velocity, as was the case in the present study.

R1.C19: In general, the discussion lacks depth. We do not clearly identify the main results of the study and the application that these results could have. Since the results part is a bit difficult to understand (although the statistical analysis are relevant), a clear and detailed discussion (with objective clues) is needed. I am further surprised by the small number of references that are used in the discussion. The development of a web application is an interested feature of this article, and should be more highlighted in my opining rather that only cited in the last sentence of the conclusion.

Response: Thank you for sharing your concerns and providing valuable suggestions to improve the manuscript. We agree that both the practical relevance of the present findings and the web application should be emphasized in the discussion section. In particular, our findings on the robustness of strength-endurance models should be highlighted, as this provides a novel perspective that, to our knowledge, has not been discussed in research yet.

Regarding the small number of references, we decided to focus any comparisons of our results to studies that featured the barbell bench press exercise, preferably studies that standardized loads to the individual 1-RM. Based on the reviewer’s suggestion, we added another reference to our discussion that investigated the reliability of the NFL-225 test, which can be regarded as a rep max test using a constant load. Other studies we found on the reliability of RTF were deemed not suitable for comparison. For example, Hoeger et al. only investigated the reliability of RTF performed on resistance training machines of a 16-station Universal Gym apparatus over the course of a pilot study without specifying the time interval in between test-retest trials and other essential methodological specifications that are potentially crucial to the reproducibility of a test [14]. 

Before: […] the 1-RM was found to be likely less than the minimal load increment […]

After: Line 282: […] the 1-RM was found to be likely less than the smallest load increment […]

Added sentences: Line 320-333: Other studies investigated the reproducibility of RTF in absolute loads. For example, Mann et al. analyzed the test-retest reliability of NCAA Division I football players in the NFL-225 test, which is a repetition maximum test using a fixed load of 225 lbs or 102.3 kg in the bench press exercise [22]. The authors reported an ICC of 0.98 to 0.99 and a typical error of 1.0 to 1.3 repetitions across three trials, the typical error corresponding to what has been calculated as SEM in the present study. While it is difficult to evaluate at what percentage of the 1-RM each participant performed the NFL-225 test in the absence of a 1-RM test, the authors estimated it to be around 67.9% 1-RM for athletes with a body mass below 100.5 kg and around 44.6% 1-RM for heavier athletes. Therefore, the majority of participants performed the NFL-225 test at lower relative loads compared to the present study. Given this fact, the reports of Mann et al. [22] correspond well to the results of the present study (SEM for RTF at 70% 1-RM [90% HDI] = 1.1 [0.8, 1.4] repetitions), especially when considering the large between-subject variance reported by the authors, which may have contributed to the large ICC, as discussed before.

Before: […] both resulted in a high probability for Δb to fall within the threshold […]

After: Line 357-358: […] both resulted in a high probability for Δb (i.e., the change in slope and curvature parameters, respectively) to fall within the threshold […]

Added sentences: Line 373-380: Based on the findings of the present study, a freely available web application was developed using the R package shiny (version 1.7.1). The application provides practitioners with a user-friendly interface to enter data from repetition maximum tests and offers different algorithms to compute the individual and exercise-specific strength-endurance profile. Upon computation, it offers a graphical display of the profile, a model equation and an adjusted R² estimate to evaluate model fit. Furthermore, it produces an individual repetition-maximum table based on the estimated model parameters that predicts loads for a wider spectrum of RTF. A link to the web application is provided at the end of this article.

Before: To help practitioners with calculating a robust strength-endurance profile, a free-to-use web application was developed based on the findings of the present study, which can be accessed using the following link:

After: Line 403-404: To derive a robust strength-endurance profile, practitioners can access a free-to-use web application using the following link:

Response to reviewer 2

R2: I’d like to congratulate the authors on a simple yet elegant, and rigorously conducted, study. I actually have very little to suggest here and think that it could largely be published as is. The app is very nicely put together too. I just make a few comments below which the authors might wish to consider.

Many thanks

James Steele

Response: Thank you very much for these kind words and your valuable input. We hope to have addressed all points according to your expectations. 

Comments:

R2.C1: “Failure” – you use the phrase “volitional failure” and do not provide a definition for this. You might wish to consider the following article from our group that discusses definitions of terms in relation to this - https://pubmed.ncbi.nlm.nih.gov/28044366/

Response: Thank you for pointing out unclear terminology that could be potentially misinterpreted by readers. Our original understanding of “volitional failure” was the point at which participants could not complete another repetition across the full range of motion using volitional contraction of target muscles despite using maximal effort. We acknowledge that this interpretation differs substantially from the definition provided in the article referenced by the reviewer and could therefore be misunderstood by readers and contribute to terminological ambiguity in the literature. We therefore changed the expression to “momentary failure” (MF) throughout the manuscript and referred readers to its definition in the article provided by the reviewer. 

Line 27: volitional failure → momentary failure

Line 29: volitional failure → momentary failure

Line 239: volitional failure → momentary failure

Line 244, Table 1 (caption): volitional failure → momentary failure

Line 399: volitional failure → momentary failure

Added sentence: Line 158-161: A repetition maximum test was terminated once the participant was unable to complete another repetition across the full range of motion despite using maximal effort, suggesting that the point of momentary failure had been reached [1]. 

R2.C2: In a supplementary analysis for a recent meta-analysis from our group (https://sportrxiv.org/index.php/server/preprint/view/109/version/120), we collated data from some studies (https://osf.io/td26u/) reporting group level results for repetitions performed to failure at different relative loads. We did explore group level strength-endurance profiles (though in order to compare self-selected repetitions numbers to what could be performed; https://osf.io/xqz9a/). Anyway, I just thought it might be of interest considering this current study. We only fit a simple linear model to it, but it would be interesting to see how well the other models you describe might fit.

Response: We would like to thank the reviewer for sharing these valuable findings from his research group. Indeed, the scatter plot showing pooled meta-analytic data in https://osf.io/xqz9a/ suggests that a curvilinear model might provide a good fit to the bivariate relationship, especially when considering loads below 60% 1-RM. It would be interesting to see if an exponential or hyperbolic model provides a better fit for the data in the Meta-Analysis. There are various sources supporting the idea of the relationship being curvilinear. We therefore decided to include the topic in our discussion.

Added sentences: Line 364-372: To decide which of the two models to apply in a practical setting, practitioners should also consider statistical qualities other than the robustness of models. For example, both the model fit und predictive validity can be considered essential characteristics of a valuable strength-endurance profile. While previous research provided some evidence that the relationship may be considered approximately linear at high loads [5, 9, 15, 23], it has been suggested that the relationship actually follows a curvilinear trend when considering the full spectrum of loads [10, 15, 16]. Therefore, practitioners might want to resort to applying the 2-parameters exponential regression rather than the linear regression to model strength-endurance profiles, as research has not proposed any explicit disadvantages reasoning against its use. 

R2.C3: I appreciate the reasoning for not randomising the loads, though think it might be worth mentioning that this is a possible limitation that could in and of itself introduce some degree of systematic bias. Perhaps just mention it in the discussion.

Response: Thank you for sharing your concerns. We agree that the presence of systematic bias resulting from an order effect cannot be ruled out. In fact, our laboratory recently started data acquisition for a replication study to analyze the magnitude of systematic bias resulting from a non-randomized single-visit protocol (as described in the present manuscript) compared to a multiple-visit approach where single RM tests are executed on different days in randomized order (similar to what has been applied by other researchers, e.g. reference 17, 18). The potential limitations of our methodological design were communicated to readers as suggested by the reviewer. 

Added sentences: Line 381-386: It should be pointed out that the order of repetition maximum tests was not randomized in the present study. Hence, a possible systematic effect of the earlier sets performed to momentary failure on subsequent sets and, thus, the presence of systematic bias in the RTF performed cannot be excluded. Future research should strive to compare different test protocols and identify a valid, yet practically applicable approach to acquiring the necessary data for model computation. 

R2.C4: I might also add to the statistical analysis when describing the models for strength-endurance profiles that the random effects for participants included both intercepts and slopes. It is clear from the equations, but not all are mathematically inclined and so explicitly mentioning this in the text might be worthwhile.

Response: The authors would like to thank the reviewer for this valuable suggestion. It was quite challenging to decide which aspects of the statistical analysis to include in the main manuscript and which ones to store as supplemental material to achieve good balance between complexity of information and transparency. The authors agree it should be clarified in the main manuscript that all model parameters were free to vary across participants. However, we are not sure whether the expressions “slope” and “intercept” might be appropriately reflecting parameters for non-linear models, especially for the 3-parameters exponential regression (Ex3) and the critical load model (Crit). As suggested by Morton et al. [24], the intercept of Crit is not expressed as a single parameter in the original model, but can be accessed through reparameterization using all three model parameters CL, ALC (=L’) and k. Similarly, the intercept in Ex3 is not expressed as a single parameter, but the resulting sum of a + c. We therefore believe it might be better to use unspecific terminology and address them as “model parameters”.

Deleted: using subjects as a random effect

Added sentence: Line 210-211: Importantly, all of the abovementioned parameters were modeled as random effects that were free to vary across subjects. 

R2.C5: The small systematic increase in 1RM, and perhaps RTFs, might be explained by the test practice effect as Jeremy Loenneke’s group have discussed (e.g., https://pubmed.ncbi.nlm.nih.gov/27875635/, https://pubmed.ncbi.nlm.nih.gov/28463902/)

Response: Thank you for providing further resources to complement the discussion of our findings. We addressed the potential explanation in the discussion as suggested by the reviewer.

Added sentence: Line 338-347: A systematic increase in the 1-RM between test and retest has previously been described on numerous occasions for various exercises [25]. Interestingly, Ribeiro and colleagues reported that this time effect did not interact significantly with participants’ experience in resistance training [26]. While the magnitude of the systematic change (Δt [90% HDI] = 1.9 kg [1.0, 2.7]) could be considered trivial in the present study, given the smallest load increment was 2.5 kg, previous research suggested that the effect may occur over the course of multiple consecutive retest trials as a result of practicing the test [26–28]. Similarly, the time effect of RTF performed at 90%, 80% and 70%-1RM showed a high probability for being less than 1 repetition. Despite the RTF at 80% and 70% 1-RM indicating a systematic difference between T1 and T2, the magnitude of this effect is likely trivial. 

R2.C6: I think it would be worthwhile to include, similarly to the 1RM/RTF table, a table showing the parameter estimates from each model for T1 and T2 in addition to the change parameter estimated. It would be nice for example to compare to estimates from other studies (I appreciate the data are available so a reader could do this themselves if they wanted too though).

Response: The authors would like to thank the reviewer for his suggestion. We agree that readers might want to compare parameter estimates to those of other studies and, therefore, they should be provided with absolute values in addition to relative and standardized parameters. In particular, this might help readers to understand, why an interpretation of standardized change effects in the present study should absolutely be complimented by an analysis of relative change effects. As suggested by the reviewer, a table was added (Table 2), summarizing posterior predictive distributions of absolute parameter values at T1 and T2, as well as absolute change effects. Consequently, the former Table 2 (standardized and relative change effects) was renumbered to Table 3.

Added sentences: Line 261-262: Table 2. Summary of posterior predictive distributions of absolute parameter values during test (T1) and retest (T2)

Before: Table 2 

After: Line 259 / Line 362: Table 3

Before: Posterior predictive distributions for […]

After: Line 246-247: Posterior predictive distributions of subject-level model parameters at T1 and T2 are summarized in Table 2. Moreover, posterior predictive distributions for […]

R2.C7: Lastly, you mention the mean-variance relationship in the discussion. I just thought it worth highlighting that this is very much apparent for repetitions performed, particularly for their log transformation (see meta-analytic estimate from the supplementary data in our meta-analysis mentioned: https://osf.io/fznhu?show=view&view_only=).

Response: Thank you for sharing these insights with us. It would be interesting to see if this relationship differs by exercise (or a similar factor like agonist muscle group) to some extent, in the sense that some exercise categories tend to result in less between-subject variance of RM across loads. We are more than welcome to further exchange views with the reviewer independently of the content of the present article.

There is indeed some evidence to support this relationship, also when looking into single studies who tested multiple loads or repetition maxima. As indicated by reviewer 2 and reviewer 1, we expanded upon the mean-variance relationship in the discussion to strengthen our point. The topic was addressed as part of our response to comment #16 of reviewer 1.

References

1. Steele J, Fisher J, Giessing J, Gentil P. Clarity in reporting terminology and definitions of set endpoints in resistance training. Muscle Nerve 2017; 56(3):368–74.

2. Suchomel TJ, Nimphius S, Bellon CR, Hornsby WG, Stone MH. Training for Muscular Strength: Methods for Monitoring and Adjusting Training Intensity. Sports Med 2021; 51(10):2051–66.

3. Hackett DA, Cobley SP, Davies TB, Michael SW, Halaki M. Accuracy in Estimating Repetitions to Failure During Resistance Exercise. J Strength Cond Res 2017; 31(8):2162–8.

4. García-Ramos A, Torrejón A, Feriche B, Morales-Artacho AJ, Pérez-Castilla A, Padial P et al. Prediction of the Maximum Number of Repetitions and Repetitions in Reserve From Barbell Velocity. Int J Sports Physiol Perform 2018; 13(3):353–9.

5. Brechue WF, Mayhew JL. Upper-body work capacity and 1RM prediction are unaltered by increasing muscular strength in college football players. J Strength Cond Res 2009; 23(9):2477–86.

6. Ware JS, Clemens CT, Mayhew JL, Johnston TJ. Muscular endurance repetitions to predict bench press and squat strength in college football players. J Strength Cond Res 1995; 9(2):99–103.

7. LeSuer DA, McCormick JH, Mayhew JL, Wasserstein RL, Arnold MD. The Accuracy of Prediction Equations for Estimating 1-RM Performance in the Bench Press, Squat, and Deadlift. J Strength Cond Res 1997; 11(4):211–3.

8. Wood TM, Maddalozzo GF, Harter RA. Accuracy of Seven Equations for Predicting 1-RM Performance of Apparently Healthy, Sedentary Older Adults. Meas Phys Educ Exerc Sci 2002; 6(2):67–94.

9. Reynolds JM, Gordon TJ, Robergs RA. Prediction of One Repetition Maximum Strength from Multiple Repetition Maximum Testing and Anthropometry. J Strength Cond Res 2006; 20(3):584–92.

10. Mayhew JL, Johnson BD, Lamonte MJ, Lauber D, Kemmler W. Accuracy of prediction equations for determining one repetition maximum bench press in women before and after resistance training. J Strength Cond Res 2008; 22(5):1570–7.

11. Hughes LJ, Banyard HG, Dempsey AR, Peiffer JJ, Scott BR. Using Load-Velocity Relationships to Quantify Training-Induced Fatigue. J Strength Cond Res 2019; 33(3):762–73.

12. Bergstrom HC, Dinyer TK, Succi PJ, Voskuil CC, Housh TJ. Applications of the Critical Power Model to Dynamic Constant External Resistance Exercise: A Brief Review of the Critical Load Test. Sports (Basel) 2021; 9(2):15.

13. Richens B, Cleather DJ. The relationship between the number of repetitions performed at given intensities is different in endurance and strength trained athletes. Biol Sport 2014; 31(2):157–61.

14. Hoeger WW, Hopkins DR, Barette SL, Hale DF. Relationship between repetitions and selected percentages of one repetition maximum: A comparison between un-trained and trained males and females. J Strength Cond Res 1990; 4(2):47–54.

15. Desgorces FD, Berthelot G, Dietrich G, Testa MSA. Local muscular endurance and prediction of 1 repetition maximum for bench in 4 athletic populations. J Strength Cond Res 2010; 24(2):394–400.

16. Sakamoto A, Sinclair PJ. Effect of movement velocity on the relationship between training load and the number of repetitions of bench press. J Strength Cond Res 2006; 20(3):523–7.

17. Moss AC, Dinyer TK, Abel MG, Bergstrom HC. Methodological Considerations for the Determination of the Critical Load for the Deadlift. J Strength Cond Res 2021; 35(Suppl 1):S31-S37.

18. Dinyer TK, Byrd MT, Vesotsky AN, Succi PJ, Bergstrom HC. Applying the Critical Power Model to a Full-Body Resistance-Training Movement. Int J Sports Physiol Perform 2019; 14(10):1364–70.

19. Anders JPV, Keller JL, Smith CM, Hill EC, Housh TJ, Schmidt RJ et al. The Effects of Asparagus Racemosus Supplementation Plus 8 Weeks of Resistance Training on Muscular Strength and Endurance. J Funct Morphol Kinesiol 2020; 5(1):4.

20. LaChance PF, Hortobagyi T. Influence of Cadence on Muscular Performance During Push-up and Pull-up Exercise. J Strength Cond Res 1994; 8(2):76–9.

21. Fountain WA, Valenti ZJ, Lynch CE, Guarnera SR, Meister BM, Carlini NA et al. Order of concentric and eccentric muscle actions affects metabolic responses. J Sports Med Phys Fitness 2021; 61(12):1587–95.

22. Mann JB, Ivey PJ, Brechue WF, Mayhew JL. Reliability and smallest worthwhile difference of the NFL-225 test in NCAA Division I football players. J Strength Cond Res 2014; 28(5):1427–32.

23. Brzycki M. Strength Testing - Predicting a One-Rep Max from Reps-to-Fatigue. JOPERD 1993; 64(1):88–90.

24. Morton RH, Redstone MD, Laing DJ. The Critical Power Concept and Bench Press: Modeling 1RM and Repetitions to Failure. Int J Exerc Sci 2014; 7(2):152–60.

25. Grgic J, Lazinica B, Schoenfeld BJ, Pedisic Z. Test-Retest Reliability of the One-Repetition Maximum (1RM) Strength Assessment: a Systematic Review. Sports Med Open 2020; 6(1):31.

26. Ribeiro AS, do Nascimento MA, Mayhew JL, Ritti-Dias RM, Avelar A, Okano AH et al. Reliability of 1RM test in detrained men with previous resistance training experience. IES 2014; 22(2):137–43.

27. Mattocks KT, Buckner SL, Jessee MB, Dankel SJ, Mouser JG, Loenneke JP. Practicing the Test Produces Strength Equivalent to Higher Volume Training. Med Sci Sports Exerc 2017; 49(9):1945–54.

28. Dankel SJ, Counts BR, Barnett BE, Buckner SL, Abe T, Loenneke JP. Muscle adaptations following 21 consecutive days of strength test familiarization compared with traditional training. Muscle Nerve 2017; 56(2):307–14.

---

## [Decision Letter · Decision Letter 1]

22 Apr 2022

Reproducibility of strength performance and strength-endurance profiles: a test-retest study

PONE-D-22-01272R1

Dear Mr Benedikt Mitter

We’re pleased to inform you that your manuscript has been judged scientifically suitable for publication and will be formally accepted for publication once it meets all outstanding technical requirements.

Kind regards,

Mathieu Gruet, Ph.D

Academic Editor

PLOS ONE

Additional Editor Comments (optional):

Reviewers' comments:

Reviewer's Responses to Questions

**Comments to the Author**

1. If the authors have adequately addressed your comments raised in a previous round of review and you feel that this manuscript is now acceptable for publication, you may indicate that here to bypass the “Comments to the Author” section, enter your conflict of interest statement in the “Confidential to Editor” section, and submit your "Accept" recommendation.

Reviewer #1: All comments have been addressed

Reviewer #2: All comments have been addressed

2. Is the manuscript technically sound, and do the data support the conclusions?

Reviewer #1: Yes

Reviewer #2: Yes

3. Has the statistical analysis been performed appropriately and rigorously? 

Reviewer #1: Yes

Reviewer #2: Yes

4. Have the authors made all data underlying the findings in their manuscript fully available?

Reviewer #1: Yes

Reviewer #2: Yes

5. Is the manuscript presented in an intelligible fashion and written in standard English?

Reviewer #1: Yes

Reviewer #2: Yes

6. Review Comments to the Author

Reviewer #1: I have no further comments, the authors have made a substantial revision work; congratulation on that

Reviewer #2: Thank you for your edits and responses. I have no further comments to add - again, great work on this piece!

7. PLOS authors have the option to publish the peer review history of their article (what does this mean?). If published, this will include your full peer review and any attached files.

Reviewer #1: **Yes: **Robin Souron

Reviewer #2: **Yes: **James Steele

---

## [Editor Report · Acceptance letter]

27 Apr 2022

PONE-D-22-01272R1 

Reproducibility of strength performance and strength-endurance profiles: a test-retest study 

Dear Dr. Mitter:

I'm pleased to inform you that your manuscript has been deemed suitable for publication in PLOS ONE. Congratulations! Your manuscript is now with our production department. 

Kind regards, 

on behalf of

Dr. Mathieu Gruet 

Academic Editor

PLOS ONE